# Molecular Regulation of α3β4 Nicotinic Acetylcholine Receptors by Lupeol in Cardiovascular System

**DOI:** 10.3390/ijms21124329

**Published:** 2020-06-18

**Authors:** Sanung Eom, Chaelin Kim, Hye Duck Yeom, Jaeeun Lee, Shinhui Lee, Yeong-Bin Baek, Jinseong Na, Sang-Ik Park, Gye-Yeop Kim, Chang-Min Lee, Jun-Ho Lee

**Affiliations:** 1Department of Biotechnology, Chonnam National University, Gwangju 61886, Korea; yeomself2355@gmail.com (S.E.); chaelinkim0215@gmail.com (C.K.); hyeduck@gmail.com (H.D.Y.); jaeeun3023@gmail.com (J.L.); dltlstn39@gmail.com (S.L.) lovespinwheel@gmail.com (J.N.); 2College of Veterinary Medicine, Chonnam National University, Gwangju 61186, Korea; chakswo@nate.com (Y.-B.B.); sipark@chonnam.ac.kr (S.-I.P.); 3Department of Physical Therapy, Dongshin University, Naju 58245, Korea; kykim@dsu.ac.kr

**Keywords:** nicotinic acetylcholine receptor, lupeol, cardiovascular system, triterpenoid

## Abstract

Cardiovascular disease (CVD) occurs globally and has a high mortality rate. The highest risk factor for developing CVD is high blood pressure. Currently, natural products are emerging for the treatment of hypertension to avoid the side effects of drugs. Among existing natural products, lupeol is known to be effective against hypertension in animal experiments. However, there exists no study regarding the molecular physiological evidence against the effects of lupeol. Consequently, we investigated the interaction of lupeol with α3β4 nicotinic acetylcholine receptors (nAChRs). In this study, we performed a two-electrode voltage-clamp technique to investigate the effect of lupeol on the α3β4 nicotine acetylcholine receptor using the oocytes of *Xenopus laevis*. Coapplication of acetylcholine and lupeol inhibited the activity of α3β4 nAChRs in a concentration-dependent, voltage-independent, and reversible manner. We also conducted a mutational experiment to investigate the influence of residues of the α3 and β4 subunits on lupeol binding with nAChRs. Double mutants of α3β4 (I37A/N132A), nAChRs significantly attenuated the inhibitory effects of lupeol compared to wild-type α3β4 nAChRs. A characteristic of α3β4 nAChRs is their effect on transmission in the cardiac sympathetic ganglion. Overall, it is hypothesized that lupeol lowers hypertension by mediating its effects on α3β4 nAChRs. The interaction between lupeol and α3β4 nAChRs provides evidence against its effect on hypertension at the molecular-cell level. In conclusion, the inhibitory effect of lupeol is proposed as a novel therapeutic approach involving the antihypertensive targeting of α3β4 nAChRs. Furthermore, it is proposed that the molecular basis of the interaction between lupeol and α3β4 nAChRs would be helpful in cardiac-pharmacology research and therapeutics.

## 1. Introduction

Cardiovascular disease (CVD) is a major global cause of mortality and of global interest due to the increasing trend of cardiovascular risk factors. Cardiovascular risk factors include hypertension, dyslipidemia, being overweight, obesity, and diabetes. Among them, hypertension is the strongest or one of the strongest risk factors for almost all different types of CVD [1,2,3]. Hypertension refers to a condition in which high blood pressure is maintained for the long term. Blood pressure (BP) is unconsciously controlled by an increase through the cardiac sympathetic nervous system and a decrease through the cardiac parasympathetic nervous system.

Nicotinic acetylcholine receptors (nAChRs) mediate cardiac ganglion transmission in the autonomic nervous system concerning CVD [4,5]. These nAChRs are composed of five protein subunits [6] that form a functional homogeneous receptor, such as α7, α9, and α10. The α2–α6 and β2–β4 subunits combine to form a functional heteromeric receptor in the organs and nervous-system regions [7]. The activation of nAChRs in association with acetylcholine is capable of causing biological reactions (excitatory postsynaptic potentials) by allowing cations such as Na^+^ and Ca^2+^ to flow into the intracellular region when the channel is opened [8]. The α3β2 and α3β4 expression types are mainly located in the cardiac autonomic nervous system, and they are involved in controlling BP [5,9,10].

Among pentacyclic triterpenoids, lupeol is the most widely known subgroup, and it is a natural product globally consumed by humans as a dietary agent in the form of fruits, medicinal plants, and vegetables [11,12]. Lupeol exerts enormous pharmacological effects on the treatment of chronic vascular diseases such as hyperlipidemia, high blood pressure, cancer, and antiviral and anti-inflammatory conditions [13,14,15]. Remarkably, lupeol is also effective against hypertension and obesity, which are the highest risk factors for CVD [16]. To date, various clinical and preclinical experiments have been conducted, but the molecular and physiological mechanisms underlying the interaction between lupeol and α3β4 nAChRs are unknown.

In the present study, we investigated the effect of lupeol on the α3β4 nicotinic acetylcholine receptor. To confirm the electrophysiological effects of lupeol, α3β4 nACh RNA was injected into oocytes, and inward peak currents were recorded using a two-electrode voltage clamp of oocytes expressing α3β4 nAChRs. Furthermore, we investigated whether lupeol competitively binds to acetylcholine-binding sites. It was observed that lupeol inhibited the activity of α3β4 nAChRs in a concentration-dependent, voltage-independent, noncompetitive, and reversible manner.

## 2. Materials and Methods

### 2.1. Materials

cDNAs for the bovine α3 and β4 subunits of the neuronal nAChRs were obtained from a neurobiotechnology laboratory (Chonnam National University, Republic of Korea). Figure 1 shows the structure of lupeol (PubChem CID: 259846). Lupeol was dissolved in dimethyl sulfoxide (DMSO) before use; the stock solution was diluted with ND96 buffer and used. The final concentration of DMSO was less than 0.05%. All other reagents used in this study were obtained from Sigma (St. Louis, MO, USA).

### 2.2. In Vitro Transcription

The cDNAs of bovine nACh α3 and β4 were linearized with SalⅠ and transcribed using an in vitro transcription kit with T7 polymerase. The synthesized RNA was dissolved in nucleotide-free water, and the mixture was divided into aliquots of 1 μg/μL final concentration and stored at −80 °C until use.

### 2.3. Molecular-Docking Studies

Molecular-docking studies were carried out using an Intel core i7, 2.20 GHs PC with 16 GB RAM running a Windows 10 64-bit operating system using Autodock Tools (version 1.5.6) by the Scripps Research Institute (La Jolla, CA, USA). The protein structure of α3β4 nAChRs was obtained from the Protein Data Bank (ID code 5T90), and the 3D structure of the ligand (lupeol) was obtained from Pubchem. The protein–ligand complex was programmed using AutoDock Tools and considered with minimized binding energy, inhibition constant, and intermolecular energy. The complex was analyzed using Ligplot (ver. 4.5.3) by EMBL-EBI and Pymol (ver. 1.8.4.2) by Schrödinger. Ligplot showed interactions between protein and ligand. Pymol was used to measure the distance between complex and the mutagenesis of amino acids of α3β4 nAChRs.

### 2.4. Mutagenesis

The DNA of nACh receptors α3 and β4 used for point mutations was obtained by PCR with Pfu DNA polymerase from a Quikchange II Site-Directed Mutagenesis Kit. To perform the point mutation, we first made the primer fit the condition. The conditions included primer length, GC content, template temperature, GC clamp (GC-rich 3 ′end), and avoiding a repeat sequence. The chimeric primer was amplified by PCR to generate point-mutated receptor subunits. Dpn I was added to remove the methylated and less mature parental cDNA. The point-mutated receptor DNA was transformed into a DH5 α-competent cell that was the insertion site of the DNA into the pGEM vector. All DNA generated by PCR was sequenced to confirm the presence of the mutation by sending it to Cosmogentech Cop.

### 2.5. Oocyte Preparation and RNA Injection

The frogs used in the experiment, *Xenopus laevis*, were managed following the high standards of Chonnam National University institution guidelines (CNU IACUC-YB-2016-07, July 2016). After frogs were anesthetized in ice, oocytes were removed and made available in 2 cultures per frog. Oocytes were separated by shaker for two hours in OR2 buffer (82.5 mM NaCl, 2 mM KCl, 1 mM MgCl_2_, and 5 mM HEPES, pH 7.5) with collagenase. Stage V–VI oocytes were selected and maintained for 2 to 4 days at 18 °C in ND96 buffer (96 mM NaCl, 2 mM KCl, 1 mM MgCl_2_, 1.8 mM CaCl_2_, and 5 mM HEPES, pH 7.5) with 0.1% penicillin and streptomycin. Individual oocytes were injected at a ratio of 1:1 in 40 ng of mRNA using a nanoinjector (Drummond Scientific, Broomall, PA, USA).

### 2.6. Data Recordings Using Two-Electrode Voltage Clamp

Recordings were performed after 2 days of injecting RNA into oocytes. A single oocyte was placed in a net chamber in continuous flow (flow rate: 2 mL/min) of ND96 buffer. Each oocyte was pierced with two electrodes filled with 3M KCl with resistances of 0.2 MΩ. Electrophysiological experiments were performed at room temperature using a Two-Electrode Oocyte Clamp (OC-725C; Warner Instruments, Hamden, CT, USA) Digidata 1550 series. The membrane potential of oocytes was maintained at a holding potential of −80 mV; then, 100 μM Ach and 30 μM lupeol were dissolved in ND96, and treated with 2 mL for 1 min. In the current–voltage relationship, voltage was applied from −100 to +60 mV for the injected oocytes. More details on the two-electrode voltage clamp can be found in [17,18,19].

### 2.7. Data Analysis

All experiment data were expressed as means ± standard error means (SEMs). Concentration–response currents for acetylcholine and lupeol were plotted by Sigma plot and fitted to the Hill equation using Origin software 8.0 (OriginLab, Northampton, MA, USA). The equation formula was y = V_max_ x [x]_n_/([k]_n_ + [x]_n_), where V_max_ is maximal currents, x is the concentration of acetylcholine or lupeol, k is the concentration of lupeol that corresponded to half the maximal current, and n is the Hill coefficient. The data difference between control and cotreatment of lupeol was statistically analyzed using paired and unpaired Student’s *t*-tests. T-test values of *p* < 0.05 were considered to be statistically significant.

## 3. Results

### 3.1. Effect of Natural-Product Lupeol on I_ACh_

In the two-electrode voltage-clamp experiments, when 100 μM of acetylcholine was employed instead of the ND96 buffer, an inward current was generated in the oocytes injected with α3β4 nAChR RNA. This outcome confirmed the expression of a functional receptor in the oocytes. Treatments with lupeol and mecamylamine (MEC) alone showed no change in current, but cotreatment of lupeol (30 μM) with acetylcholine (ACh) of 100 μM significantly inhibited I_ACh_ compared with the treatment of ACh (100 μM) alone, like official antagonist MEC (Figure 1; *n* = 4–5 from three different frogs). The inhibition percentage of I_ACh_ was 97.9% ± 2.0% and 64.5% ± 1.8% for MEC and lupeol, respectively (Figure 1B). The inhibition of I_ACh_ by lupeol was reversible (Figure 1C). There was no effect on the I_ACh_ when lupeol was intracellularly injected without passing through α3β4 nAChRs.

### 3.2. ACh Concentration–Response Profiles and Current–Voltage Relationship

The concentration–response profiles of different concentrations of lupeol with acetylcholine revealed that cotreatments inhibited the expression of α3β4 nAChRs in a concentration-dependent manner (Figure 2A,B). The degree of inhibition induced by lupeol was calculated from the average of the peak inward current elicited after cotreatment with lupeol. The curve was fitted according to equation y = V_max_ * [lupeol]_n_/([k]_n_ + [lupeol]_n_), where V_max_ is the maximal current, k is the concentration of lupeol that corresponds to half the maximal current, and [lupeol] is the concentration of lupeol. Half-inhibitory concentration IC_50_ for lupeol was 15.4 μM (*n* = 10–13 from five different frogs). The Hill coefficient for lupeol was 1.17 ± 0.24. These data showed that lupeol exhibited inhibitory effects on nAChRs in a concentration-dependent manner.

The current–voltage relationship was demonstrated by lupeol by inhibiting the expression of α3β4 nicotine acetylcholine receptors. The experiment was performed by applying voltages from −100 to +60 mV while holding the membrane potential at −80 mV. Treatment with acetylcholine or cotreatment with lupeol showed a reversible potential close to 0 mV (*n* = 7–9 from four different frogs (Figure 2C). This means that lupeol had a binding site at the pore as an open channel blocker, or was docked in the outer vestibule. The modulation of lupeol occurred regardless of transmembrane-potential voltage.

### 3.3. Noncompetitive Inhibition by Lupeol

To study the mechanisms by which lupeol inhibits the activity (I_Ach_) of α3β4 nicotinic acetylcholine receptors, we analyzed I_ACh_ by treatments with 30 μM lupeol with different concentrations of ACh. The curves produced by the cotreatment of appropriate concentrations were similar in shape to the control, and the only difference was in the degree of inhibited current (*n* = 10–13 from five different frogs (Figure 2D)). However, cotreatment with lupeol significantly inhibited I_ACh_, elicited by 100 and 300 μM of ACh. Analysis of the protein levels of α3β4 nACh receptors with little difference between individuals of *Xenopus laevis* oocytes was nonpharmacological, so we did not analyze protein levels. To address this issue, please refer to the highly quoted review paper of Dr. Dascal [19].

### 3.4. Docked Modeling of Lupeol and α3β4 nACh Receptors

The α3 and β4 subunits of nAChR were mutated to investigate the relationship between receptor binding sites and lupeol. We employed covalent docking modeling to compare the wild-type and mutants of α3β4 nAChRs. Through best-fit analysis, it was possible to check the bonding position with a high probability in three dimensions (Figure 3). In Figure 3, lupeol appears to be located at the boundary between the α3 and β4 subunits. An in silico model revealed a combination with both subunits. From the best-fit docking results, it appears that lupeol formed strong hydrogen bonds with wild-type α3β4 nAChRs and not with the mutant. To find the binding pocket with the most stable binding, and to confirm the activity of each residue, programming was performed using Autodock 4.0. Lupeol interacted with four residues of α3β4 nAChRs in the wild type: α3 (I37A; distance = 3.7 and 3.4 Å), α3 (D118; distance = 3.5 and 3.8 Å), β4 (L129; distance = 3.6, 3.5, 3.3, and 3.6 Å), and β4 (N132; distance = 3.6, 3.2, and 3.5 Å) (Figure 4C). In the mutant, residues were (Figure 4D): α3 (I37A; distance = 7.9 and 7.5 Å), α3 (D118A; distance = 3.3, 3.8, and 3.2 Å), β4 (L129A; distance = 6.0, 5.5, and 5.0 Å), and β4 (N132A; distance = 3.8, 3.6, and 4.1 Å).

### 3.5. Inhibitory Effect of Lupeol on Double-Mutant α3β4 (I37A/N132A) nACh Receptors

To find out whether inhibition applies in vitro rather than in silico, a mutant gene was obtained by point mutation, and a two-electrode voltage clamp (TEVC) was used to compare the inhibitory effect of wild types and mutants in acetylcholine and lupeol. All mutant genotypes confirmed binding with lupeol using TEVC. Among the mutants, double-mutant α3β4 (I37A/N132A) showed a reduced inhibitory effect (48%) compared to one-subunit mutants α3 (I37A) + Wild β4 (66%) and Wild α3 + β4 (N132A) (65%). Additional half-inhibitory concentration (IC_50_), Hill coefficient (n_H_), and *I*_max_ values are presented in Table 1. The wild type exhibited a 90% inhibitory effect. Acetylcholine and various concentrations of lupeol were treated together to obtain data (Figure 5D). The Hill coefficient value was 1.5 ± 0.4. These results indicated that lupeol-induced regulation of α3β4 nAChR activity was closely related to the I37 residue of the α3 subunit and the N132 residue of the β4 subunit. Furthermore, it is evident that lupeol-induced regulation of α3β4 nAChR activity was closely related to the I37 residue of the α3 subunit and the N132 residue of the β4 subunit. Results obtained in silico and in vitro exhibited complete similarity.

## 4. Discussion

Lupeol has numerous pharmacological activities [15]; hence, various clinical and preclinical studies were conducted [14]. In particular, supplementation with lupeol (0.68 g/kg) for seven weeks significantly decreased BP in hypertensive rats [16]. However, studies at the molecular-cell level that lupeol is effective for hypertension are poorly understood. Herein, we uncovered evidence that lupeol lowers hypertension by characterizing the interaction of lupeol with α3β4 at the molecular-cell level. It is hypothesized that the interaction of lupeol with α3β4 nAChRs may have the potential for CVD treatment via the suppression of hypertension.

In this study, we confirmed that: (1) cotreatment with lupeol and acetylcholine inhibited I_Ach_-expressing α3β4 nAChRs in oocytes; (2) cotreatment of acetylcholine and lupeol inhibited I_Ach_-expressing α3β4 nAChRs in a reversible and concentration-dependent manner; and (3) inhibition by lupeol occurred in a voltage-independent and noncompetitive manner.

The mechanism of interaction of lupeol with the α3β4 nicotinic acetylcholine receptor has not been determined. There are several possibilities for the interaction of lupeol with α3β4 nAChRs. One possibility is that it may act as an open channel blocker of α3β4 nAChRs. As shown in Figure 2, inhibition generated by the lupeol was observed in both hyperpolarized and depolarized potentials, and it was voltage-independent. However, known open channel blockers are known to act in a voltage-dependent manner due to charges present in the oocyte’s transmembrane electrical domain.

Another possibility is that lupeol inhibits I_ACh_ by interfering with the binding of acetylcholine to the binding site of α3β4 nAChRs. However, in our experiment data, we did not show any curve movement even though we had treated ACh with various concentrations of lupeol. Accordingly, lupeol may not act as a competitive inhibitor of α3β4 nAChRs.

In previous studies, nicotine ligands were commonly known to bind to alpha and beta subunits, and form ligand-binding sites [20,21]. We investigated whether the binding of lupeol was affected by various nAChR mutants. After finding potential-candidate binding sites via a molecular-docking study, all candidates were point-mutated. The combination of the three mutants, α3β4 (I37A/L129A), α3β4 (K124D/L129A), and α3β4 (D118R/N132I), showed slight differences from lupeol to the wild-type receptor. Most remaining mutants had similar effects on the wild-type α3β4 nAChRs, and were either not or weakly expressed, and were not effective. Overall data in this study suggested that one position in nAChRs plays an important role in the binding of lupeol: I37A of the α3 subunit and N132A of the β4 subunit. However, since only a portion of nAChR has been mutagenized and experimented with, it is not possible to exclude the possibility that other sites may be important.

α3β4 and α3β2 nAChRs are mainly expressed in the autonomic nervous system [22] according to results that looked at combination types that are mainly expressed in vivo in cardiac-sympathetic and - parasympathetic neurons, respectively. Subunit α3 is an essential element in cardiac autonomic neurons. The expression of α3β2 nAChRs was confirmed in cardiac-parasympathetic neurons but no significant result was seen with α3β4 nAChRs. Therefore, the inhibitory effect of lupeol and α3β4 nAChRs on cardiac-sympathetic neurons led to a decrease in norepinephrine release by a reduction in the postsynapse signal, resulting in BP decrease.

In summary, lupeol inhibited I_ACh_ in receptor-expressing oocytes, thereby demonstrating that it functions in a concentration–response, noncompetitive, and voltage-independent manner. The inhibitory effect of lupeol can be utilized as a novel therapeutic approach for the antihypertensive targeting of α3β4 nAChRs. Furthermore, residues at specific positions in the α3β4 (I37A/N132A) subunits play a crucial role in the binding of lupeol to the receptor. It is hypothesized that the present understanding of the interaction between lupeol and α3β4 nAChRs should be helpful in pharmacological CVD studies.

## Figures and Tables

**Figure 1 ijms-21-04329-f001:**
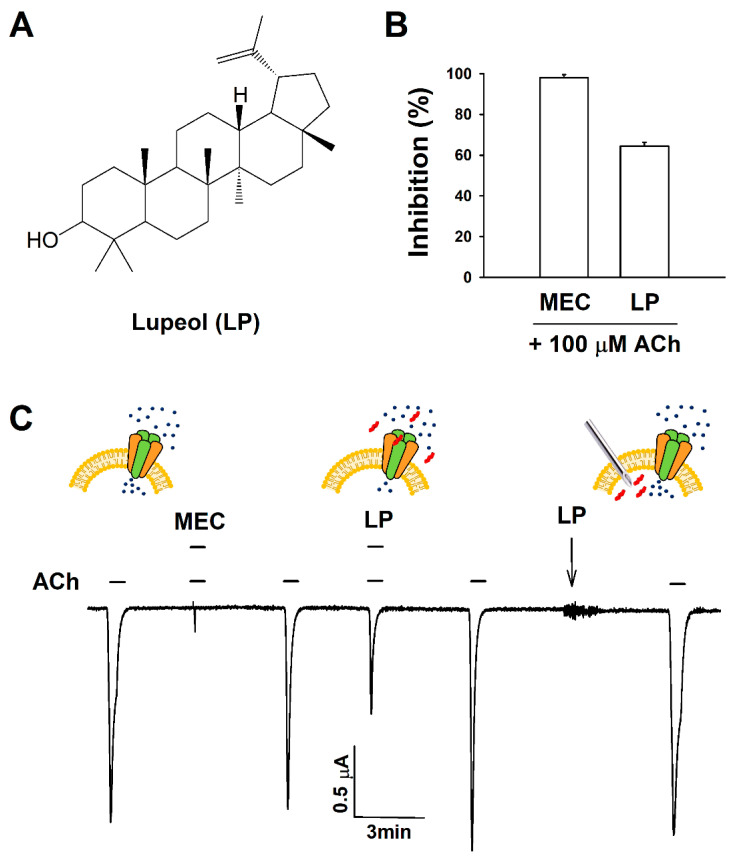
Structure and regulatory effects of lupeol on the α3β4 nicotinic acetylcholine receptors. (**A**) Chemical structure of lupeol (LP). (**B**) Summary of inhibitory effects of cotreatment of lupeol with acetylcholine. (**C**) Typically, 100 μM acetylcholine was applied with or without 30 μM lupeol. Mecamylamine (MEC) is an antagonist of nicotinic acetylcholine receptors (nAChRs). MEC shown at concentration of 10 μM. Arrow, point were LP was not treated onto oocyte surface but injected into cells. Traces representative of 5–9 separate oocytes from 3–5 different frogs. I_Ach_ recorded at holding potential of −80 mV before lupeol treatment. Small blue circle, cation; red one, lupeol.

**Figure 2 ijms-21-04329-f002:**
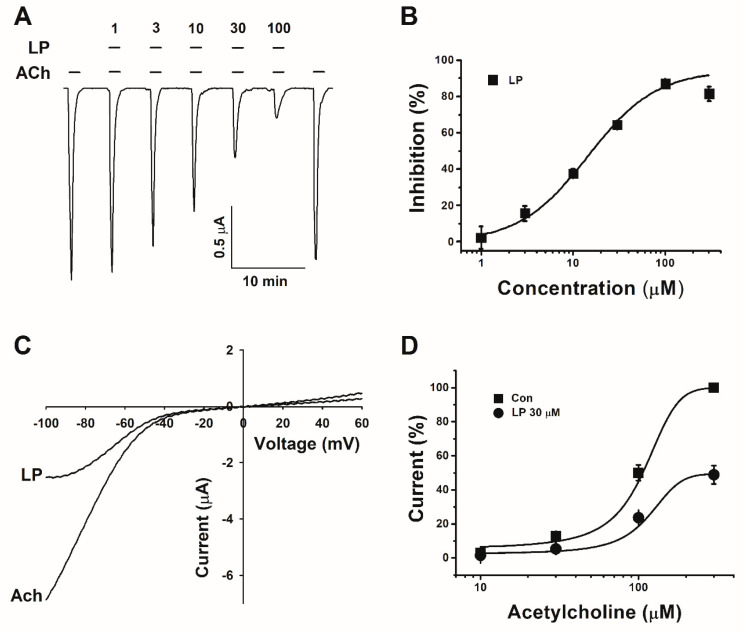
Mechanism by which lupeol interacts with α3β4 nicotinic acetylcholine receptors. (**A**) Inward currents of trace for cotreatment of acetylcholine and lupeol in α3β4 nACh receptor. (**B**) Concentration–response relationship induced by cotreatment of lupeol in α3β4 nACh receptors. Each point represents mean ± SEM (*n* = 10–13/group). (**C**) Representative current–voltage relationship obtained by using voltage ramps from −100 to +60 mV at holding potential of −80 mV. Voltage steps treated with 100 μM acetylcholine alone, and cotreated with 30 μM lupeol with ACh (*n* = 7–9 from four different frogs). (**D**) I_Ach_ induced by various concentrations of acetylcholine (■) and cotreatment with 30 μM lupeol (●). Oocytes voltage-clamped at holding potential of −80 mV. Each point represents mean ± SEM (*n* = 10–13 from five different frogs).

**Figure 3 ijms-21-04329-f003:**
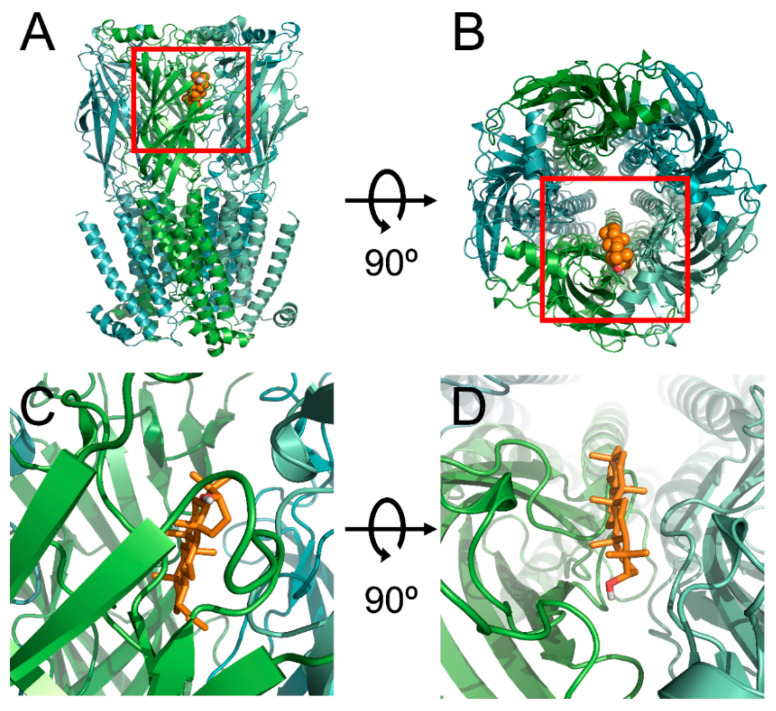
Computational molecular modeling of lupeol docked to α3β4 nicotinic acetylcholine receptor. (**A**,**C**) Side views of docked lupeol in complex with nACh α3β4 receptor. (**B**,**D**) Top view of docking model.

**Figure 4 ijms-21-04329-f004:**
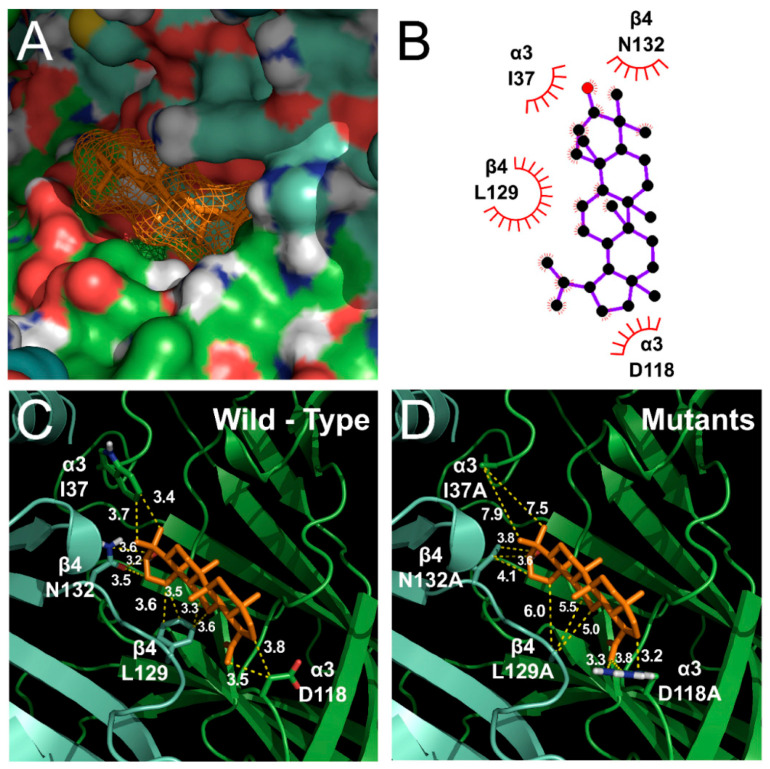
Binding pocket view and docking results comparing wild type and mutant in lupeol docked to α3β4 nAChRs. (**A**) Lupeol located in binding pocket in extracellular area between Segments 1 and 2 of α3β4 nicotinic acetylcholine receptors. (**B**) Two-dimensional schematic presentation of predicted binding mode of lupeol in ligand-binding pocket. Ligands and important residues shown. (**C**,**D**) Binding interface and lupeol of wild type. (**C**) Four mutant channels in which mutations disturbed interaction of lupeol to varying degrees.

**Figure 5 ijms-21-04329-f005:**
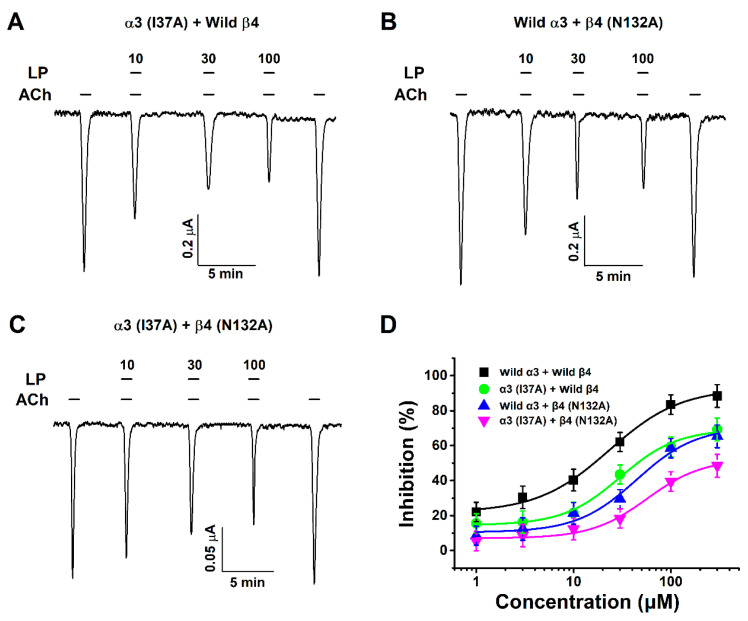
Effect of lupeol on double-mutant α3β4 nicotinic acetylcholine receptors. (**A**–**C**) Inward currents of concentration–response relationship to coapplication of acetylcholine and lupeol with 10, 30, and 100 μM concentrations of oocytes expressing α3 (I37A) + Wild β4, Wild α3 + β4 (N132A), and α3β4 (I37A/N132A) nAChRs. (**D**) Concentration–response graphs showing effect of different concentrations of lupeol on mutant α3β4 nACh receptors in presence of 100 μM acetylcholine. Each point is mean ± SEM (*n* = 6–7 from three different frogs). Additional half-inhibitory concentration, Hill coefficient, and *I*_max_ values described in Table 1.

**Table 1 ijms-21-04329-t001:** Effects of lupeol on subunit mutants of α3β4 nicotinic acetylcholine receptors.

Subunit Mutants	*I* _max_	IC_50_	n_H_
Wild α3 + Wild β4	92.8 ± 4.6	23.2 ± 3.7	1.2 ± 0.2
α3 (I37A) + Wild β4	69.4 ± 4.0	30.1 ± 5.0	1.5 ± 0.4
Wild α3 + β4 (N132A)	70.7 ± 9.3	44.6 ± 14.7	1.4 ± 0.6
α3 (I37A) + β4 (N132A)	52.3 ± 5.2	57.0 ± 13.2	1.5 ± 0.4

Values represent means ± SEM (*n* = 6−7/group). Currents elicited at holding potential of −80 mV. IC_50_ (μM), Hill’s coefficient, and *I*_max_ (%)determined as described in Materials and Methods.

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
