# Peer review of "Molecular Regulation of α3β4 Nicotinic Acetylcholine Receptors by Lupeol in Cardiovascular System"

_ijms, 2020, doi:10.3390/ijms21124329_

Round 1

Reviewer 1 Report

The authors reported the molecular interaction between lupeol and α3β4 nAChRs and they provides evidence against hypertension by regulatory effects on α3β4 nAChRs. They showed inhibitory effect of lupeol is proposed as a novel therapeutic cellular approach of cardiovascular disease
Overall, this is a well-done study but some minor changes would enhance the manuscript.

Minor points
1. I would like to suggest that author double-check the spacing word and misspelling through the manuscript.
2. The author should be accurately indicated the concentration of MEC used in page 3.
3. It will be better information if the author provides the Pubchem CID number of the lupeol.
4. Please add the flow rate at line 119 in page 4.
5. "Treatments with only lupeol and mecamylamine (MEC) exhibited no effect, but co-treatment with lupeol (30 μM) and 100 μM acetylcholine (ACh) yielded significantly lower IACh compared with the treatment with acetylcholine alone" This sentence at line 140 in page 5 is inappropriate.
6. The author omitted explanation that was no intracellular regulation of lupeol in figure 1C.
7. Tested oocytes numbers are different in figure 5 legend and table 1 legend.

Author Response

1# Reviewer’s comments

The authors reported the molecular interaction between lupeol and α3β4 nAChRs and they provide evidence against hypertension by regulatory effects on α3β4 nAChRs. They showed inhibitory effect of lupeol is proposed as a novel therapeutic cellular approach of cardiovascular disease

Overall, this is a well-done study but some minor changes would enhance the manuscript.

Authors’ response:

Thank you for considerate review. We revised the manuscript in a better way, thanks to your kindness which comments on the details. We hope this revised manuscript will meet your expectations. All revised parts highlighted in red. The reference style has also been changed for the International journal of molecular sciences.

1# Reviewer’s Minor points

  1. I would like to suggest that author double-check the spacing word and misspelling through the manuscript.

Authors’ response:

All authors of this paper double-checked the misspelling and deleted unnecessary spacing (For example, double-spacing page 2, lines 58). If we need further correction of the manuscript, we plan to entrust a professional English editing company once more.

  1. The author should be accurately indicated the concentration of MEC used in page 3.

Authors’ response:

We added the concentration of MEC and highlighted in red at page 3, line 83.

  1. It will be better information if the author provides the Pubchem CID number of the lupeol.

Authors’ response:

We actively agree with your thoughts, so we added Pubchem CID number of the lupeol at page 2, line 75.

  1. Please add the flow rate at line 119 in page 4.

Authors’ response:

We added the flow rate at page 4, line 123.

  1. "Treatments with only lupeol and mecamylamine (MEC) exhibited no effect, but co-treatment with lupeol (30 μM) and 100 μM acetylcholine (ACh) yielded significantly lower IACh compared with the treatment with acetylcholine alone" This sentence at line 140 in page 5 is inappropriate.

Authors’ response:

We corrected them at page 5, lines 145-147.

  1. The author omitted explanation that was no intracellular regulation of lupeol in figure 1C.

Authors’ response:

We also agree entirely with the lack of explanation. So, we added the explanation to the section of Figure 1(page 3, line 83-85) and section of the results (page 5, line 150-151).

  1. Tested oocytes numbers are different in figure 5 legend and table 1 legend.

Authors’ response:

Thank you for finding the missing parts when we checked. We corrected “7-8” to “6-7” at page 9, line 239.

Reviewer 2 Report

This study looks at the efect of lupeol in relation to CVD. There is value in looking into natural extracts/compounds to prevent and treat CVD however I find difficult to get the essence of the paper. I would suggest that the authors review the phrasing and the organising of the manuscript. I have highlighted and commented directly on the manuscript - this is not exhaustive. Below is a list of further points that require attention:

  • Figure 1: it doesn't make sense to have it in the Methods
  • No indication of treatment with lupeol in the Methods (concwentration, length etc.)
  • I am confused between "expression" of the receptors and "activity" on figure 2. I don't believe protein levels have been analysed, but only the level of response of the receptors.
  • Table 1: no units.

You may want to look at the conclusion to enrich it with some references to support some of your statements. 

Author Response

2# Reviewer’s comments

This study looks at the efect of lupeol in relation to CVD. There is value in looking into natural extracts/compounds to prevent and treat CVD however I find difficult to get the essence of the paper. I would suggest that the authors review the phrasing and the organising of the manuscript. I have highlighted and commented directly on the manuscript - this is not exhaustive. Below is a list of further points that require attention:

Authors’ response:

First of all, thank you for your suggestion we tried to change the parts you kindly highlight, but we couldn't find them in both files (docx and pdf) and emails. We asked the editor, but we couldn't get the highlighted file. Preferentially, we corrected comments (further points) which are your thorough review and salient observations. All revised parts highlighted in red.

-Figure 1: it doesn't make sense to have it in the Methods

Authors’ response:

We added the explanation of Figure 1 to Methods section of Data recordings (page 4, line 127-128), the section of Figure 1(page 3, line 83-85) and section of the results (page 5, line 150-151). References are cited for detailed explanation (page 4, line 129-130).

-No indication of treatment with lupeol in the Methods (concwentration, length etc.)

Authors’ response:

We added lupeol concentration and treatment time at page 4, line 127-128. It also serves to solve the lacking of understanding of Figure 1 in the methods mentioned above.

-I am confused between "expression" of the receptors and "activity" on figure 2. I don't believe protein levels have been analysed, but only the level of response of the receptors.

Authors’ response:

Frist of all, we are sorry to use the term "expression" that is confusing to the readers you mentioned above. We replaced from "expression" to "activity (IAch)" at page 6, line 181.

You raised an important point, because this comment is one of the main questions asked by non-electrophysiology majors. In electrophysiology studies using Xenopus oocytes, measuring protein expression levels of neuronal receptors is considered medically meaningless. This is because if they are the same gene, there is no significant difference in protein expression between individuals Xenopus laevis oocytes. Further, the graph analyzing the activity of the receptor according to the concentration of the drug is pharmacologically significant. So we added sentences for easier reading and cite a review paper “The use of Xenopus oocytes for the study of ion channel” by Dr. Dascal who authority of this field at page 6, line 186-189.

-Table 1: no units.

Authors’ response:

We added units to table section at page 9, line 239-240.

-You may want to look at the conclusion to enrich it with some references to support some of your statements.

Authors’ response:

We deleted all but one of the three references that are essential one reference(Siddique and Saleem, 2011) at page 10, line 244.

Round 2

Reviewer 2 Report

Thank you for amending your paper and I am sorry you couldn't see the comments on the document. I appreciate that your response to the comments which enhance your manuscript.

On the new version, the sentence on line 18 & 19 (abstract) doesn't make sense and it should be rephrased - I know what you want to say but the sentence does not work.

On line 60, you state 'such as obesity": the way teh sentence is written, it would appear that you classify obesity as a chronic vascular disease when I believe it is a condition that causes chronic vascular disease - it should therefore be changed.

Line 62 - it should be "To date" and not "Until date".

Table 1 title: I would remove "number of" as it is confusing.

Line 276: you state "and experimented": should it not be "and experimented with" or "and experimented on"?

Finally, you mention your institutional guidelines regarding the Xenopus, I am sure this include any ethical approval of your study so it would be good to state as the study was approved by the ethical committee, to avoid any misunderstanding.

Author Response

Thank you for amending your paper and I am sorry you couldn't see the comments on the document. I appreciate that your response to the comments which enhance your manuscript.

Authors’ response:

Thank you for your considerate comments and appreciate for giving us the excellent comments to improve our manuscript.

Comment #1

On the new version, the sentence on line 18 & 19 (abstract) doesn't make sense and it should be rephrased - I know what you want to say but the sentence does not work.

Authors’ response:

We corrected the first sentence in the abstract section to “Cardiovascular disease (CVD) occurs worldwide and has a high mortality rate, and the highest risk factor for developing CVD is high blood pressure.” (page 1, line 18-19)

Comment #2

On line 60, you state 'such as obesity": the way teh sentence is written, it would appear that you classify obesity as a chronic vascular disease when I believe it is a condition that causes chronic vascular disease - it should therefore be changed.

Authors’ response:

We agree with your thorough insight so, we deleted “obesity” at page 2, line 60.

Comment #3

Line 62 - it should be "To date" and not "Until date".

Authors’ response:

We changed from “Until date” to "To date" at page 2, line 62.

Comment #4

Table 1 title: I would remove "number of" as it is confusing.

Authors’ response:

We deleted “number of” in Table 1 of the title (page 9, line 239). Thank you for your in-depth observation.

Comment #5

Line 276: you state "and experimented": should it not be "and experimented with" or "and experimented on"?

Authors’ response:

We agree with your suggestion so, we corrected from "and experimented” to "and experimented with" at page 10, line 276.

Comment #6

Finally, you mention your institutional guidelines regarding the Xenopus, I am sure this include any ethical approval of your study so it would be good to state as the study was approved by the ethical committee, to avoid any misunderstanding.

Authors’ response:

We added the ethical approval at page 4, line 115. I am sorry for the mistake and thank you for finding this important part.
